# Understand KRAS and the Quest for Anti-Cancer Drugs

**DOI:** 10.3390/cells10040842

**Published:** 2021-04-08

**Authors:** Chang Woo Han, Mi Suk Jeong, Se Bok Jang

**Affiliations:** 1Institute of Systems Biology, Pusan National University, Jangjeon-dong, Geumjeong-gu, Busan 46241, Korea; hotorses@pusan.ac.kr; 2Institute for Plastic Information and Energy Materials and Sustainable Utilization of Photovoltaic Energy Research Center, Pusan National University, Jangjeon-dong, Geumjeong-gu, Busan 46241, Korea; 3Department of Molecular Biology, College of Natural Sciences, Pusan National University, Jangjeon-dong, Geumjeong-gu, Busan 46241, Korea

**Keywords:** KRAS, cancer, signaling pathway, inhibitors

## Abstract

The KRAS oncogene is mutated in approximately ~30% of human cancers, and the targeting of KRAS has long been highlighted in many studies. Nevertheless, attempts to target KRAS directly have been ineffective. This review provides an overview of the structure of KRAS and its characteristic signaling pathways. Additionally, we examine the problems associated with currently available KRAS inhibitors and discuss promising avenues for drug development.

## 1. Introduction

*RAS* genes (*HRAS*, *KRAS*, and *NRAS*) encode 21 kDa monomeric small guanosine triphosphate (GTP)-hydrolases (GTPases) that function as regulators of intracellular signaling transduction cascades involved in cell growth, differentiation, and survival. Normally, the cycling of RAS protein by GTPases is tightly regulated by the binding of guanosine triphosphate (GTP) to nucleotide exchange factors (GEFs) and by the hydrolysis of GTP to guanosine diphosphate (GDP) by GTPase activating proteins (GAPs) [1,2]. KRAS is frequently mutated in pancreatic, colorectal, endometrial, biliary tract, lung, and cervical cancer. For example, oncogenic missense mutations in the KRAS gene frequently occur at glycine (Gly) 12 or 13 or glutamine (Gln) 61 codons located on the P loop and Switch II region and are related to the impairment of stable nucleotide-binding in its active site. These mutations impair GTPase cycling and result in accumulations of bound and active GTP, which result in activation of multiple downstream effectors [3,4]. In this review, we examine aberrant activated KRAS signaling in tumors and place focus on promising KRAS inhibitors.

## 2. Structure

KRAS is a 21 kDa monomeric, membrane-localized GTPase with a conserved domain, known as the G domain (residues 1–165, Figure 1A), and a less conserved C-terminal tail, called the hypervariable domain (residues 166–188, Figure 1A) [5]. KRAS consists of five α2-helixes (α1–α5) and six β-sheets (β1–β6). GTPase has two switches— switches I (residues 30–38) and II (residues 60–76)—which undergo major conformational changes in response to changes in guanine nucleotide (GTP to GDP or vice versa) [6]. The phosphate-binding loop (P-loop, residues 10–17) of KRAS participates in the induction of the conformational change that occurs during its GDP-bound (inactive) and the GTP-bound (active) states (Figure 1B). The γ-phosphate of GTP interacts with tyrosine (Tyr) 32 and threonine (Thr) 35 residues of the switch I region. Additionally, conserved Gly 60 in the switch Ⅱ region interacts with γ-phosphate [7]. KRAS is the most frequently mutated proto-oncogene in human cancers, and mutated KRAS remains locked in an active state, relaying uncontrolled proliferative signals. Jonathan M. Ostrem et al. identified that oncogenic mutations result in conformational change and functional activation of RAS proteins by impairing both intrinsic GTPase activity and GAP-stimulated GTP hydrolysis [6]. Recently, many studies have focused on attempts that target exchange nucleotide. Recent studies have focused on attempts to block nucleotide exchange [8,9]. Several structure-based inhibitors (e.g., SML-8-73-1, H-REV107 peptide, BAY-293, or BI-3406) have been reported to target the guanine nucleotide binding site of the KRAS mutant and inhibit GDP to GTP exchange [10,11,12,13]. The Cys-A-A-X C-terminal motif, where A is isoleucine, leucine, or valine, and X is either methionine or serine, is located in the KRAS hypervariable region (HVR). KRAS trafficking on the plasma membrane plays essential roles in multiple signaling pathways that control various processes, and the Cys-A-A-X C-terminal motif of KRAS undergoes several post-translational modifications (PTMs) that facilitate its trafficking to the inner surface of the plasma membrane [14,15,16].

(i) KRAS is synthesized as a biologically inactive cytosolic propeptide (Pro-Ras). (ii) The first modification of this propeptide is catalyzed by farnesyltransferase (FTase) and involves covalent addition of the 15-carbon farnesyl group (C-15) from farnesyl diphosphate (FDP) onto the cysteine residue of the CAAX sequence. (iii) Next, the -AAX sequence is cleaved by a CAAX protease RAS converting CAAX endopeptidase 1 also known as RAS converting enzyme 1 (RCE1), and this is followed by methylation of the carboxyl-prenylated cysteine residue through isoprenylcysteine carboxyl methyltransferase (ICMT). (iv) Finally, palmitoyl transferase (PTase) catalyzes the addition of palmitic acid (a 16-carbon fatty acid) to upstream cysteine residues.

These KRAS modifications enhance protein hydrophobicity and plasma membrane anchoring, and membrane-anchored KRAS proteins cycle between the GTP-bound active form and the GDP-bound inactive form. The associated conformational changes of KRAS serve as master moderators of a large number of signaling pathways involved in various cellular processes.

## 3. Oncogenic KRAS Signaling Pathways in Human Cancer

### 3.1. GEFs and GAPs

KRAS-associated signaling pathways are persistently activated in many cancers, where they participate in cellular growth and proliferation, differentiation, protein synthesis, glucose metabolism, cell survival, and inflammation (Figure 2) [3]. KRAS is activated by GEFs: SOS1 and SOS2 (two sons of sevenless 1 and 2), GRB2 (growth factor receptor-bound protein 2), RASGRF2 (Ras protein-specific guanine nucleotide releasing factor 2), or RasGRPs (Ras guanine nucleotide releasing proteins). GEFs are activated by transmembrane receptors—G protein-coupled receptors (GPCRs) and receptor tyrosine kinases (RTKs), which mediate important cellular responses including proliferation, differentiation, and survival [17,18,19,20]. GEFs consist of the Ras exchange motif (REM), which is involved in the stabilization of binding to Ras, and a CDC25 homology catalytic domain (CDC25H); both functional domains are necessary for the nucleotide exchange activity of GTPase. The CDC25H catalytic domain interacts with the GDP-bound inactive state of KRAS via an interface involving the Switch I and Switch II regions, and this precedes the formation of GEFs-Ras-GDP complex. Subsequently, GEF catalyzes the release of GDP and consists of nucleotide-free Ras-GEF complex state. When GDP bound to RAS is released by GEF, KRAS is replaced by GTP (rather than GDP) in a process driven by elevated intracellular concentrations of GTP [21,22,23]. KRAS is inactivated by GAPs, such as neurofibromatosis type 1 (NF1), Ras protein activator like -1 or -2 (RASAL1 or 2), and disabled homolog 2-interaction protein (DAB2IP). KRAS has weak intrinsic GTP hydrolysis activity, which is greatly enhanced by GAPs. The carboxyl-terminus of GAPs contain RAS GTPase-activating (RASGAP) domain that catalyzes the activation of the GTPase cycle by hydrolyzing the “active” GTP-bound form of KRAS to produce the “inactive” GDP-bound form [24,25,26,27].

Oncogenic KRAS missense mutations prevent GTP hydrolysis and result in the accumulation of KRAS in the active state, which causes persistent downstream signaling.

### 3.2. Raf-MEPK-ERK Pathway

The Raf/mitogen-activated protein kinase (MEK or MAPK)/extracellular-signal-regulated kinase (ERK) pathway is a well-characterized KRAS pathway that leads to cell growth, cell death inhibition, cell cycle progression, invasiveness, and the induction of angiogenesis [28]. Raf is a member of a family of serine/threonine (Ser/Thr) kinases, which include A-Raf, B-Raf, and C-Raf/Baf1. Raf is recruited to the plasma membrane through binding to the switch I domain of membrane-anchored active RAS and by lipid binding. Mutated KRAS induces ERK activation and tumor progression by activating the formation of RAF homo- or heterodimers [29]. Subsequently, Raf stimulates mitogen-activated protein kinases 1 and 2 (MEK1 and MEK2), which ultimately activate the downstream extracellular signal-regulated kinases 1 and 2 (ERK1 and ERK2) [30,31]. MEK1 and MEK2 belong to the family of dual-specificity kinases (DSKs) that phosphorylate Tyr and Ser/Thr residues within the activation loop of their MAP kinase substrates, and these phosphorylations are dependent on and activated by the phosphorylations of serine 218 and 222 in their activation segments by RAF [32]. Activated MEKs directly interact with ERK through their N-terminal regions and catalyze the phosphorylations of Thr and Tyr residues on the TEY (Thr-Glu-Tyr) motif of the activation loop of ERK to activate ERK [33]. Phosphorylated ERK then translocates to the nucleus, where it phosphorylates and activates a variety of transcriptional factors, including ternary complex factor (TCF) Elk-1 and serum response factor accessory proteins Sap-1a, Ets1, c-Myc, and Tal, and these activations upregulate the expressions of the immediate-early protein c-Fos, which enables cell cycle progression through G0/G1 mitogenic signals [34,35].

### 3.3. PI3K-Akt-mTOR Pathway

Phosphatidylinositol 3-kinases (PI3Ks) are crucial for signaling downstream of mutant KRAS-driven tumors as they regulate cell growth, cell cycle entry, cell survival, cytoskeleton, reorganization, and metabolism. PI3K isoforms are divided into three classes based on their structural and biochemical characterization. Class I PI3Ks are heterodimers that contain one of four catalytic p110 subunits (p110α (PI3CA), p110β (PIK3CB), p110γ (PIK3CG), and p110δ (PIK3CD)) and a regulatory subunit (p85α, p85β, p55γ, p101, and p84). Class II PI3Ks occur in three isoforms (PI3K-C2α (PIK3C2A), PI3K-C2β (PIK3C2B), and PI3K-C2γ (PIK3C2G)). Class III PI3Ks have a catalytic subunit (vacuolar sorting protein 34 (Vps34)) and a regulatory subunit (Vps15) [36,37]. PI3K family members are activated through RTKs or GPCRs on the plasma membrane and generate phosphatidylinositol-3,4,5-phosphate (PIP3) by phosphorylating phosphatidylinositol 4, 5-bisphosphate (PIP2) [38]. Furthermore, activated KRAS interacts and activates PI3Ks [39]. PIP3 initiates downstream signaling by recruiting serine-threonine protein kinases B (Akt) and 3-phosphoinositide-dependent protein kinase-1 (PDK1) to the cell membrane, and PDK1 phosphorylates and thereby activates Akt. AKT activation requires phosphorylation at two main activating sites—Thr 308 in the activation T-loop and Ser 473 in its C-terminal hydrophobic motif—via PDK1 and PDK2, respectively [40]. Once activated, AKT mediates cell growth and survival by phosphorylating mTOR (serine/threonine kinase mammalian target of rapamycin), which acts in two functionally distinct complexes, i.e., RAPTOR associated mTOR complex 1 (mTORC1) and RICTOR associated mTOR complex 2 (mTORC2). mTORC1 is relatively sensitive to rapamycin, while mTORC2 is resistant [41]. Recently, it was reported that mTORC2 and Akt constitute a positive feedback loop via mTORC2 subunit SIN1 (stress-activated protein kinase interacting protein 1) phosphorylation, and phosphorylation of this subunit leads to the phosphorylation of Akt serine (Ser) 473 [42]. Phosphorylation of two main activating sites Thr 308 and Ser 473 of Akt by PDK1 and mTORC2 is essential for the Akt-mediated signaling network, which includes glycogen synthase kinase-3 (GSK-3), forkhead box O (FOXO), tuberous sclerosis complex (TSC2), and mTORC1 [43,44].

### 3.4. The RalGEF-Ral Pathway

The RalGEF-Ral pathway (Ral-selective guanine nucleotide exchange factors–Ras-like pathway) is the third-best characterized effector of KRAS-dependent human oncogenesis [45]. Like Ras, Ral proteins (RalA and RalB) are small GTPases that cycle between inactive GDP-bound and active GTP-bound states. However, Ral GTPase is dependent on RALGEF and GTPase activating proteins (RALGAPs) to catalyze the GDP–GTP exchange because of its weak intrinsic GTPase activity [46]. Mutant KRAS activates RalGEFs, which leads to the formation of the active GTP-bound state of Ral GTPases. Activated Ral GTPases stimulate a wide spectrum of downstream effectors and regulate their activations. Activated Ral GTPases stimulate a wide spectrum of downstream effectors and regulate their activations; these include ZO-1–associated nucleic acid-binding protein (ZONAB), exocyst subunits Sec5/Exo84, Filamin, RalA Binding Protein 1 (also known as RalBP1, RLIP1, and RIP1), and phospholipase D1 (PLD1) [47,48,49,50,51]. Activation of Ral effector signaling has been shown to play important roles in proliferation, survival, metastasis, actin cytoskeleton reorganization, transcription regulation, intracellular membrane trafficking, and kinase cascade signaling.

## 4. Inhibitors

Oncogenic mutations at codons Gly 12, 13, and Gln 61 of KRAS occur in ~86% of KRAS-mediated human cancers, including pancreatic (90%), colon (40%), and non-small cell lung cancer (20%) [52]. However, efforts to develop inhibitors that directly target the corresponding constitutively active mutant KRAS proteins have been unsuccessful. Several potential binding sites have been identified using structure-based computational methods, but these surface sites are relatively flat without deep hydrophobic pockets [53,54]. Nevertheless, there have been several attempts to overcome this problem. General mechanisms of the drug for inhibiting KRAS are as follows: manipulating RAS in the GTP-bound state, disruption of activated KRAS-effector interactions, multimerization/stabilization of inactive KRAS, and decreasing KRAS trafficking to membrane. The GDP dissociation inhibitors (GDIs) inhibit KRAS by regulating the rate of exchange of GDP for GTP of small GTPases. GDI binds with high affinity to only the lipid-modified form of the RAS superfamily proteins and keep GTPase in the GDP-bound state [55,56]. Recent studies proved that the KRAS helices α4–α5 contribute to the formation of KRAS dimers and higher-order nanoclusters. Xiaolin Nan et al. confirmed that the KRAS G12D mutation forms a dimer and activates the MAPK pathway in BHK21 and TRex-293 cell lines [57]. In addition, several studies searched for KRAS dimer interfaces using computational techniques [58,59,60]. Based on these results, various high-affinity small molecules (e.g., NS1, DARPin K13, and BI-2852) target the α4–α5 dimerization interface for inhibitions between oncogene KRAS and downstream effectors [61,62]. One mutant-specific inhibitor targets the guanine nucleotide-binding site of mutant KRAS. This GDP analogue, SML-8-73-1 (SML), targets the guanine nucleotide-binding site of KRAS G12C mutant. Biochemical and biophysical measurements suggest that SML reacts quantitatively with KRAS G12C mutant but does not react with WT KRAS [10]. In addition, SML efficiently competes with GTP and GDP for binding to the guanine pocket of KRAS G12C mutant. Furthermore, SML reduced the ERK to Akt downstream phosphorylation ratio as compared with non-treated controls in non-small cell lung cancer (NSCLC) cell lines [63]. H-REV107 peptide is another mutant-specific inhibitor, which targets the guanine nucleotide-binding site of KRAS G12V mutant. This inhibitor was shown to interact with KRAS G12V in the GDP-bound inactive state to form a stable complex, and thus to block the activation function of KRAS. Additionally, H-REV107 peptide inhibited pancreatic and colon cancer cell lines by suppressing the phosphorylation levels of MEK/ERK.

In particular, H-REV107 peptide suppressed pancreatic tumor growth in a xenotransplanted mouse model by reducing tumor volume and weight [11]. Overall, the targeting of the guanine nucleotide-binding site of KRAS provides a developmental strategy for targeting the inhibition of KRAS mutations in cancer patients. In another study, attempts were made to selectively inhibit the KRAS–SOS1 interaction using small molecules (BAY-293 and BI-3406) mimicking an orthosteric SOS helix. BAY-293 was found to block GTP binding to KRAS and suppress the formation of KRAS-SOS1 complex, which reduced phospho-ERK activity by ∼50% in NSCLC cell lines [12]. Additionally, BI-3406 is a potent and selective SOS1-KRAS interaction inhibitor that reduces RAS-GTP levels and curtails MAPK pathway signaling in KRAS-driven cancers [13]. These inhibitors disrupt activated KRAS-effector interactions.

Another study focused on FTase, which is responsible for the posttranslational modification of the C-terminal CAAX motif. FTase inhibitors (FTIs) and geranylgeranyltransferase type I (GGTase-I) inhibitors (GGTIs) inhibited the plasma membrane association and subcellular localization of KRAS. FTIs have been developed for cancer treatment for more than twenty years, and several clinical studies have demonstrated that FTIs (antroquinonol, tipifarnib, lonafarnib, BMS-214662, and L778123) can effectively suppress tumor growth by blocking farnesylation with little toxicity [64]. However, several studies have shown that KRAS can be alternatively prenylated by GGTase-I in FTI-treated cells [65], thus countering the notion that KRAS mutation requires a combination of FTIs and GGTIs. In Colo357, A-549, and Calu-1 human cancer cell lines, KRAS prenylation was found to be inhibited by FTI-277 and GGTI-298 co-treatment [66].

Directly blocking mutant KRAS activity has proven to be extremely challenging. Recent studies have targeted KRAS downstream effector pathways [67]. The best-characterized effector pathways are the Raf-MEPK-ERK and PI3K-Akt-mTOR pathways, which initiate cascades of protein-protein interactions leading to proliferation, survival, cell cycle regulation, wound healing and tissue repair, integrin signaling, cell migration, or tumorigenesis [68]. The oral multikinase inhibitor sorafenib (Nexavar^®^) suppresses tumor growth by targeting Raf serine/threonine kinases (Raf-1, wild-type B-Raf, and oncogenic B-Raf V600E) and receptor tyrosine kinases (VEGFR-1, VEGFR-2, VEGFR-3, PDGFR-β, Flt-3, and c-Kit) in a variety of human cancers [69]. Selumetinib (AZD6244, ARRY-142886) is a selective and potent second-generation allosteric ATP-non-competitive MEK1/2 inhibitor that suppresses ERK1/2 phosphorylation [70]. Sotorasib (AMG 510), a KRAS (G12C) inhibitor developed by Amgen^®^, produced significant results in phase 1 clinical trial for non-small cell lung cancer, rapidly emerging as the next new drug for lung cancer. Sotorasib inhibits MAPK signaling and tumor growth in KRAS G12C cell lines and patient-derived xenograft models [71]. PI3K-Ark-mTOR pathway inhibitors are subdivided into Pan-PI3K inhibitors (buparlisib and copanlisib) [72,73], mTOR inhibitors (everolimus) [74], and dual PI3K/mTOR inhibitors [75,76]. However, these inhibitors have been reported to cause severe and sometimes fatal adverse effects, such as autoimmune dysfunction, opportunistic infections, skin toxicity, hypertension, and hyperglycemia [77]. Thus, while effector pathway inhibition appears to be the most promising therapeutic strategy for targeting KRAS mutant cancers, significant challenges remain.

## 5. Conclusions

Small GTPase KRAS is one of the most frequently mutated oncogenes and is mutated in more than 30% of human cancers. Despite advances in our understanding of the structure, function, and cellular signaling pathways that underlie KRAS-mutant cancer development, KRAS targeting remains a therapeutic challenge, which demands further understanding of the biological and biochemical functions of KRAS. Accumulated evidence unearthed by research studies suggests that solving previously known adverse drug reactions might facilitate the development of novel drugs with significant impacts on KRAS-driven cancers. As one method to overcome the occurrence of resistance to existing inhibitors, there is a method of maximizing treatment with “Combination Treatment” of each inhibitor. Combination therapy is already being carried out in various studies and is getting significant results [78]. Thus, we believe that communications regarding the biochemical properties and biological functions of KRAS protein will accelerate the development of more effective cancer therapies.

## Figures and Tables

**Figure 1 cells-10-00842-f001:**
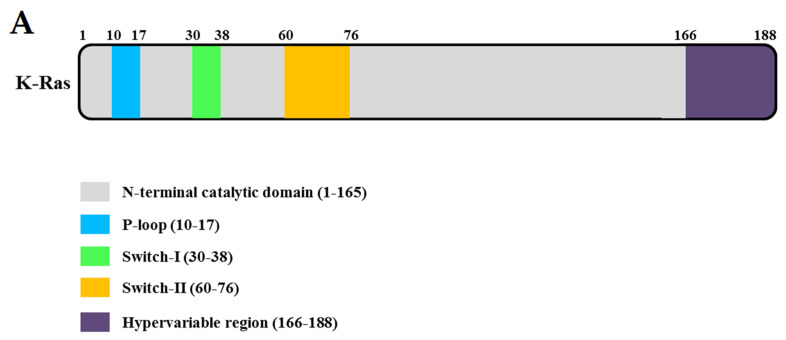
Domain structures and the three-dimensional structure. (**A**) Domain structure of full-length KRAS. (**B**) Ribbon representation of the monomeric crystal structure of KRAS (PDB ID: 7C40) using the program PyMOL. The Mg^2+^ ion is indicated by a gray circle and guanosine diphosphate (GDP) by a yellow rod. The MgGDP molecule is color-coded as follows: C yellow, O red, N blue, P purple, and Mg^2+^ gray.

**Figure 2 cells-10-00842-f002:**
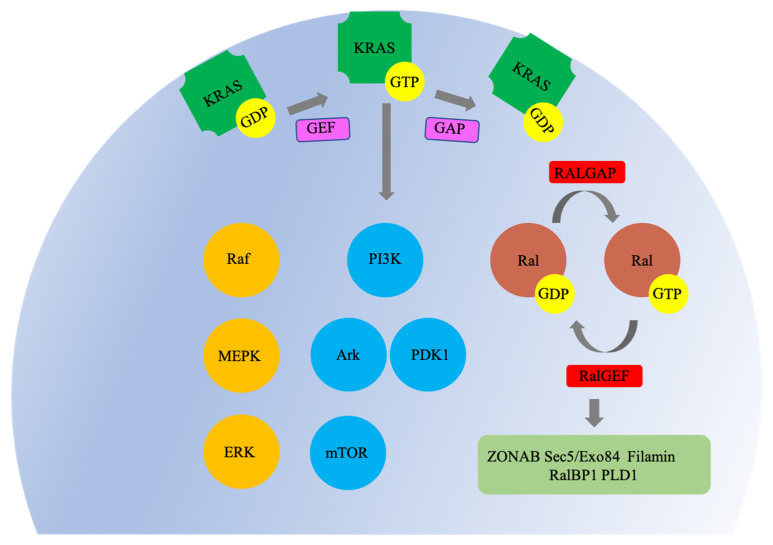
Characteristic KRAS signaling pathways. In the active guanosine triphosphate (GTP)-bound state, KRAS interacts with several families of effector proteins and stimulates their catalytic activities. Raf protein kinases activate mitogen-activated protein kinase kinases 1 and 2 (MEK1 and MEK2), which leads to extracellular-signal-regulated kinase (ERK)1/2 activation. Phosphoinositide 3-kinases (PI3Ks) generate second-messenger lipids and activate numerous target proteins, including the survival signaling kinase AkT. KRAS binding activates Ral-specific guanine nucleotide exchange factors (RalGEFs) by directing them to their Ral GTPase substrates in the plasma membrane. KRAS is indicated by a green frame shape, Raf/MEPK/ERK by an orange, PI3K/Ark/PDK1/mTOR by a sky blue, Ral by a brown, and GDP/GTP by a yellow circle. The nucleotide exchange factors (GEF) is indicated by a magenta, RALGEF and GTPase activating proteins (RALGAP)/RalGEF by a red, and ZONAB Sec5/Exo84/Filamin/RalBP1/PLD1 by a dark sea green rectangle.

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
