# Peer review of "Understand KRAS and the Quest for Anti-Cancer Drugs"

_cells, 2021, doi:10.3390/cells10040842_

Round 1

Reviewer 1 Report

Chang Woo Han and colleagues have put together a review titled “Understand KRAS and the quest for anti-cancer drugs” that address the quest for drugging the undruggable KRAS oncoprotein. Overall the topic is relevant and should get an opportunity to see the light of day, however I do have some comments that needs addressing. The authors should reorganize the section of the introduction that covers the biological basis of KRAS activity to reflect on the structural basis of KRAS inhibition. As you know, RAS activates downstream effectors through the protein–protein interaction and the only effective inhibitors would inhibit such interaction between RAS and its downstream targets. Discuss the four general mechanisms for inhibiting KRAS: manipulating RAS in the GTP-bound state, disruption of activated KRAS-effector interactions, multimerization/stabilization of inactive KRAS and decreasing KRAS trafficking to membrane.

The review discussed inhibitors of KRAS G12V and G12C. What about G12D, can it be inhibited by a similar principle?

It is known that inhibitors can have off-target and indiscriminate activity by affecting wild type KRAS resulting in side effects and toxicity. It is important (before discussing the inhibitors) to discuss the biochemical mechanism by which KRAS mutant-specific inhibitors achieve such specificity, for example by reacting with the mutant amino acid.   

The review discussed three inhibitors. This needs to be improved by classifying the inhibitors into the four categories above and discussing inhibitors that belong to each group.

What are the clinical status of these inhibitors?

Over English needs refinement some pointed out below:

Line 23: and cervical cancers.

Line 24: remove the in “the cancer-associated KRAS”

Line 25: remove the in “result in the translations of different amino acids.”

Line 30: remove the in “result in the activations of multiple downstream effectors”, change activations to activation.

Line 36: “GTPase has two switches, that is “ change to that are

Line 40: transition from to not from and

Line 47: “KRAS trafficking on the plasma membrane” trafficking to

Line 73: change elevated to activated

Line 131: removed the break at the end of the line

Line 134: “Class I PI3Ks are heterodimers that contains” change to contain

Line 136: “Class III PI3K has a catalytic subunit”  change to Class III PI3Ks

Line 141: “KRAS interacts and stimulates PI3Ks” change stimulates to activates

Line 181: “ Thus, recent studies have reported the targeting of RAS-effector interactions”. This sentence is not clear

Line 182: “binding site of KRAS mutant” change to mutant KRAS

Line 185: “mutant but does not label WT KRAS”. Change does not label WT to does not react with

Reviewer 2 Report

The authors reviewed the literature on the KRAS oncogene, which is mutated in 30% of human cancers. They described in detail the structure of KRAS mutation, the downstream signaling, and the clinical development of inhibitors for cancer patients.

Overall, the manuscript is comprehensive. However, I was left with the impression that there are some things lacking in this review.

Comments:

  • In Figure 1(B), was the ribbon representation of the monomeric crystal structure of KRAS an original work? If so, please provide details of the software used for making the ribbon representation.
  • In Figure 2, please provide specifications on the colors and their indications.
  • In the section of 3.2 (line 109), does KRAS pathway activation lead to cell growth suppression? If so, please elaborate on the mechanisms. Moreover, the authors are advised to include the negative feedback ERK activation in this section.
  • The section 4 of “Inhibitors” should be kept incomplete. The development of KRAS inhibitor is very aggressive area. Therefore, the authors should mention the promising inhibitors, published by some established journals, such as AMG-510 (Sotorasib), BI-3406 etc. Moreover, tipifarnib has been successfully used in head and neck cancer with HRAS mutation. The combination of vemurafenib and trametinib has been approved for clinical use for the cancer patients with BRAF mutation.
  • Finally, the authors should elaborate on the ideal therapeutic strategies for patients with KRAS mutation in their opinion.

Round 2

Reviewer 1 Report

The authors have made a very quick and halfhearted attempt att adressing fairly easy comments

The authors did not discuss the structural basis for KRAS inhibition as we suggested before. It will improve the quality of the review if this part is covered in the introduction.

Following comments were not addressed

We requested to discuss biochemical mechanism by which KRAS mutant-specific inhibitors achieve such specificity, for example by reacting with the mutant amino acid. However, the authors did not reply to this point. We believe this is an important point before discussing the inhibitors in this review.

Previous comment: The review discussed inhibitors of KRAS G12V and G12C. What about G12D, can it be inhibited by a similar principle?

Their response: It was added to the text (Page 6, line 7).

However, this cannot be found in the manuscript. Please discuss G12D inhibitors as well.

There are still English mistakes.

Line 27: “which result in the activation”. Change to “which result in activation”

************************

Line 48: “facilitate its trafficking on the inner surface of”. The authors did not change “on” to “to”.

************************

Line 102: ”The KRAS is indicated by a green frame shape”. The sentence should not start with “The”

************************

Line 30: remove “the” in “result in the activations of multiple downstream effectors”,

change activations to activation.

Activations modified to activation.

However, “the” was not removed.

************************

Line 40: transition from to not from and

It has been modified to transition from.

This has not been corrected.

************************

Line 47: “KRAS trafficking on the plasma membrane” trafficking to

It has been modified to trafficking on.

This has not been corrected.

************************

Reviewer 2 Report

I agree the authors responses.

Round 3

Reviewer 1 Report

No further comments beyond editorial english typo checking